# CAN PRE-TRAINED MODELS ASSIST IN DATASET DISTILLATION?

## ABSTRACT

Dataset Distillation (DD) is a prominent technique that encapsulates knowledge from a large-scale original dataset into a small synthetic dataset for efficient training. Meanwhile, Pre-trained Models (PTMs) function as knowledge repositories, containing extensive information from the original dataset. This naturally raises a question: Can PTMs effectively transfer knowledge to synthetic datasets, guiding DD accurately? To this end, we conduct preliminary experiments, confirming the contribution of PTMs to DD. Afterwards, we systematically study different options in PTMs, including initialization parameters, model architecture, training epoch and domain knowledge, revealing that: 1) Increasing model diversity enhances the performance of synthetic datasets; 2) Sub-optimal models can also assist in DD and outperform well-trained ones in certain cases; 3) Domain-specific PTMs are not mandatory for DD, but a reasonable domain match is crucial. Finally, by selecting optimal options, we significantly improve the cross-architecture generalization over baseline DD methods. We hope our work will facilitate researchers to develop better DD techniques. Our code is available at https://anonymous.4open.science/r/DDInterpreter-0DC5.

## 1 INTRODUCTION

Dataset Distillation (DD) condenses a large-scale, real-world dataset into a small, synthetic one such that models trained on the latter yield comparable performance (Wang et al., 2018; Yu et al., 2023). Thanks to its extremely high compression ratio and strong performance, DD has become a mainstream approach for dataset compression (Lei & Tao, 2023). Numerous follow-ups have been carried out in this research line, such as designing better optimization algorithms (Nguyen et al., 2021; Cazenavette et al., 2022; Wang et al., 2022; Kim et al., 2022; Jin et al., 2022; Dong et al., 2022) as well as extending the application of DD (Zhao & Bilen, 2021; Xiong et al., 2022; Song et al., 2022; Zhou et al., 2022; Zhao & Bilen, 2023).

In current DD methods (Zhao et al., 2021; Zhao & Bilen, 2021; Wang et al., 2022; Cazenavette et al., 2023), two key steps are involved: training models and calculating the matching loss between the original dataset and synthetic dataset using these models. These two steps are performed alternately in an iterative loop to optimize the synthetic dataset (Yu et al., 2023). Essentially, the step of training models in DD can be viewed as extracting knowledge from the original dataset. Given that Pre-trained Models[1] (PTMs) are ready-made knowledge repositories that contain extensive information from the original dataset, a natural question arises: *Can pre-trained models assist in dataset distillation?*

With this question in mind, we conduct a series of experiments. Firstly, we introduce a plug-and-play loss term, *i.e.*, **CLoM** (short for **C**lassification **L**oss **o**f pre-trained **M**odel). By leveraging PTMs as supervision signals, CLoM serves as stable guidance on the optimization direction of synthetic datasets. Preliminary experiments demonstrate that PTMs indeed contribute to DD. Subsequently, we systematically study for the first time the effects of different options in PTMs on DD when utilizing PTMs as supervision signals. Specifically, we consider initialization parameters[2], model architecture, training epoch and domain knowledge. In experiments involving domain knowledge, we introduce a

---

[1]It is worth noting that "Pre-trained Models" refers to models trained on the original dataset.

[2]In this paper, "different initialization parameters" refers to the model's parameters initialized with various random seeds.

novel loss term named **CCLoM** (**C**ontrastive **C**lassification **L**oss **o**f pre-trained **M**odel) to address the issue of label mismatch in PTMs across different domains. We summarize several key findings here:

- *Model diversification.* The increased diversity of PTMs, including various initialization parameters and diverse model architectures, leads to additional performance improvements in DD to some extent.

- *Training epoch.* It's not mandatory to employ well-trained models as supervision signals. Surprisingly, using sub-optimal models instead can sometimes achieve better results.

- *Domain knowledge.* PTMs used as supervision signals do not have to be trained on the original dataset; they can also come from other datasets. Among PTMs in various domains, domain-related PTMs are preferred as accurate supervision signals.

We hope that these findings will facilitate the DD community to concentrate on PTMs and inspire the development of future algorithms. Finally, by selecting optimal options, we significantly improve the cross-architecture generalization over baseline methods. Specifically, the gains on Dataset Condensation (Zhao et al., 2021) (DC), Differentiable Siamese Augmentation (Zhao & Bilen, 2021) (DSA), and Distribution Matching (Zhao & Bilen, 2023) (DM) are +4.5%/+10.7%/+5.2%, respectively.

## 2 RELATED WORK

**Dataset Distillation.** With the unlimited growth of data, DD, as a data reduction technique, has attracted widespread attention. Previous studies have demonstrated the effectiveness and feasibility (Nguyen et al., 2021; Zhao et al., 2021; Zhao & Bilen, 2021; Cazenavette et al., 2022; Wang et al., 2022; Kim et al., 2022) of this technique and apply it to various fields like continual learning (Zhao et al., 2021; Zhou et al., 2022; Zhao & Bilen, 2023), federated learning (Xiong et al., 2022; Song et al., 2022; Liu et al., 2022; Hu et al., 2022), neural architecture search (Such et al., 2020; Zhao et al., 2021; Zhao & Bilen, 2021; 2023) and recommender system (Sachdeva et al., 2022), etc.

**Pre-trained Models.** PTMs have shown promising achievements in natural language processing, computer vision, and cross-modal fields (Radford et al., 2018; 2019; Kenton & Toutanova, 2019; Brown et al., 2020; Radford et al., 2021; OpenAI, 2023). They provide a practical solution for mitigating the data hunger and poor generalization ability in deep learning (Belkin et al., 2019; Xu et al., 2020). For example, a series of models (Simonyan & Zisserman, 2014; He et al., 2016) are trained on the large-scale visual recognition dataset such as ImageNet (Deng et al., 2009). The rich knowledge in the dataset is injected into pre-trained models through model training and implicitly encoded in model parameters. Through fine-tuning, this knowledge can be transferred to numerous downstream tasks, spanning image classification (He et al., 2016), object detection (Sermanet et al., 2013; Obaid et al., 2022), image segmentation (Iglovikov & Shvets, 2018; Kalapos & Gyires-Tóth, 2022), image generation (Bellagente et al., 2023), 3D vision (Zhang et al., 2023), etc.

Unfortunately, while PTMs have successfully benefited various downstream tasks by taking advantage of the knowledge extracted from the original dataset, DD has not effectively harnessed this knowledge. In this paper, we explore the role of PTMs in DD, paving the way for future research.

## 3 BACKGROUND AND EXPERIMENTAL SETUP

### 3.1 BACKGROUND

DD aims to synthesize small-scale surrogate datasets containing similar information to the original large-scale datasets (Wang et al., 2018; Zhao et al., 2020; Wang et al., 2022). Let $\mathcal{T} = \{(x_i, y_i)\}_{i=1}^{|\mathcal{T}|}$ be the original dataset consisted of $|\mathcal{T}|$ pairs of images and corresponding labels. The synthetic surrogate dataset is denoted as $\mathcal{S} = \{(s_i, y_i)\}_{i=1}^{|\mathcal{S}|}$, where $|\mathcal{S}| \ll |\mathcal{T}|$. Then the DD task can be formulated as follows:

$$\mathcal{S}^* = \arg\max_{\mathcal{S}} \phi(\mathcal{S}, \mathcal{T}), \tag{1}$$

where $\phi$ is a task-specific loss that varies among different DD methods. Below we introduce three representative methods with each using a different technique.

DC was proposed to match the training gradients of the synthetic data and original data (Zhao et al., 2021). Given a model with parameters $\theta$, the optimization process can be expressed as:

$$\min_{\mathcal{S}} \sigma \left( \nabla_\theta \mathcal{L}(\theta; \mathcal{S}), \nabla_\theta \mathcal{L}(\theta; \mathcal{T}) \right), \tag{2}$$

where $\mathcal{L}(\cdot; \cdot)$ and $\sigma(\cdot; \cdot)$ denote the training loss and a sum of cosine distances between the two gradients of weights associated with each output node at each layer, respectively. Building upon DC, DSA applies data augmentation techniques to further enhance performance (Zhao & Bilen, 2021). It can be formulated as follows:

$$\min_{\mathcal{S}} \sigma \left( \nabla_\theta \mathcal{L}(\mathcal{A}(\mathcal{S}, \omega^{\mathcal{S}}), \theta), \nabla_\theta \mathcal{L}(\mathcal{A}(\mathcal{T}, \omega^{\mathcal{T}}), \theta) \right), \tag{3}$$

where $\mathcal{A}$ is a family of image transformations such as cropping, color jittering and flipping that are parameterized with $\omega^{\mathcal{S}}$ and $\omega^{\mathcal{T}}$. Different from DC and DSA, DM aligns the feature distributions of the original dataset and synthetic dataset in sampled embedding spaces (Zhao & Bilen, 2023), which can be formulated as:

$$\min_{\mathcal{S}} \sigma \left( \frac{1}{|\mathcal{S}|} \sum_{i=0}^{|\mathcal{S}|} f(\theta; s_i), \frac{1}{|\mathcal{T}|} \sum_{i=0}^{|\mathcal{T}|} f(\theta; x_i) \right), \tag{4}$$

where $f(\cdot; \cdot)$ is the feature extraction function and $\sigma(\cdot; \cdot)$ represents maximum mean discrepancy (Gretton et al., 2012).

### 3.2 EXPERIMENTAL SETUP

**Datasets**: We conduct primary experiments on two publicly-available datasets: CIFAR-10 (Krizhevsky et al., 2009) and CIFAR-100 (Krizhevsky et al., 2009), which are most commonly used in DD. As for experiments related to domain knowledge, we use ImageNet-32 (Chrabaszcz et al., 2017), PathMNIST (Yang et al., 2023), ImageNette[3] and ImageFruit[4]. ImageNet-32 is a dataset composed of small images downsampled from the original ImageNet (Deng et al., 2009). PathMNIST is a biomedical dataset about colon pathology. ImageNette and ImageFruit are subsets of the 10 classes in ImageNet, respectively.

**Models**: Unless otherwise specified, staying consistent with precedent DD methods (Zhao et al., 2021; Cazenavette et al., 2022; Cui et al., 2022), we use a standard ConvNet architecture with three convolutional layers (ConvNet-3) to train and evaluate synthetic datasets. As for experiments on ImageNette and ImageFruit, we increase the number of convolutional layers to 6 (ConvNet-6), in line with Cazenavette et al. (2023).

**Training configurations**: During the evaluation of synthetic datasets, we uniformly apply DSA to preprocess all synthetic images and train the augmented images with a batch size of 256 for 1,000 epochs. We employ the SGD optimizer with an initial learning rate of 0.01, a momentum of 0.9 and a weight decay of $5 \times 10^{-4}$. The learning rate is reduced by a factor of 0.1 after every 500 epochs. As for training to obtain PTMs, we train the model (without DSA) for 150 epochs, with a weight decay of $5 \times 10^{-3}$ and reduce the learning rate by 0.1 after every 50 epochs. To mitigate the impact of randomness, we repeat each experiment 5 times and report the mean and standard deviation. All experiments are conducted on a server equipped with 2 NVIDIA Tesla A100 GPUs.

## 4 CAN PRE-TRAINED MODELS ASSIST IN DATASET DISTILLATION?

PTMs have already learned valid feature representations and rich semantic information from the dataset on which they are trained (Tang et al., 2020; Gou et al., 2021). As a result, they inherently possess a certain level of knowledge about the original dataset. Therefore, utilizing PTMs as supervision signals may provide stable guidance for the optimization direction of synthetic datasets. In order to verify our conjecture, we conduct a preliminary experiment.

To leverage the knowledge embedded in PTMs for enhancing DD, we propose a plug-and-play loss term, called **C**lassification **L**oss **o**f pre-trained **M**odel (CLoM). CLoM can be portable to any existing

---

[3]ImageNette is downloaded from `https://github.com/fastai/imagenette`.

[4]ImageFruit is downloaded using the label index provided in Cazenavette et al. (2023).

Table 1: Performance comparison to different DD methods. C10 and C100 denote CIFAR10 and CIFAR100, respectively. $\mathcal{N}_m$ represents the number of PTMs with different initialization parameters (different random seeds). $\mathcal{N}_a$ denotes the number of different model architectures. Experiments with $\mathcal{N}_m = 1$ and $\mathcal{N}_a = 1$ verify whether PTMs are beneficial for DD; Experiments with $\mathcal{N}_m = 10$ and $\mathcal{N}_a = 1$ investigate the influence of initialization parameters on synthetic datasets; Experiments with $\mathcal{N}_m = 1$ and $\mathcal{N}_a = 4$ investigate the influence of model architecture on synthetic datasets.

| Dataset | Method | CLoM | $\mathcal{N}_m$ | $\mathcal{N}_a$ | IPC 10 | IPC 50 | Dataset | Method | CLoM | $\mathcal{N}_m$ | $\mathcal{N}_a$ | IPC 10 | IPC 50 |
|---|---|---|---|---|---|---|---|---|---|---|---|---|---|
| C10 | DC | ✗ | - | - | $51.4_{\pm0.3}$ | $57.5_{\pm0.2}$ | C100 | DC | ✗ | - | - | $28.7_{\pm0.3}$ | $31.4_{\pm0.4}$ |
| | | ✓ | 1 | 1 | $51.5_{\pm0.3}$ | $57.8_{\pm0.3}$ | | | ✓ | 1 | 1 | $29.4_{\pm0.4}$ | $33.7_{\pm0.4}$ |
| | | ✓ | 10 | 1 | $52.0_{\pm0.3}$ | $60.2_{\pm0.3}$ | | | ✓ | 10 | 1 | $31.8_{\pm0.3}$ | $39.5_{\pm0.4}$ |
| | | ✓ | 1 | 4 | $52.2_{\pm0.1}$ | $59.7_{\pm0.6}$ | | | ✓ | 1 | 4 | $30.2_{\pm0.2}$ | $36.0_{\pm0.6}$ |
| | DSA | ✗ | - | - | $53.3_{\pm0.2}$ | $60.7_{\pm0.5}$ | | DSA | ✗ | - | - | $32.8_{\pm0.4}$ | $42.8_{\pm0.4}$ |
| | | ✓ | 1 | 1 | $53.4_{\pm0.3}$ | $61.9_{\pm0.3}$ | | | ✓ | 1 | 1 | $33.8_{\pm0.2}$ | $43.0_{\pm0.4}$ |
| | | ✓ | 10 | 1 | $54.4_{\pm0.4}$ | $65.1_{\pm0.5}$ | | | ✓ | 10 | 1 | $35.5_{\pm0.2}$ | $43.2_{\pm0.3}$ |
| | | ✓ | 1 | 4 | $53.5_{\pm0.4}$ | $63.0_{\pm0.2}$ | | | ✓ | 1 | 4 | $34.1_{\pm0.2}$ | $43.2_{\pm0.2}$ |
| | DM | ✗ | - | - | $49.6_{\pm0.3}$ | $62.5_{\pm0.7}$ | | DM | ✗ | - | - | $30.0_{\pm0.2}$ | $43.2_{\pm0.4}$ |
| | | ✓ | 1 | 1 | $50.0_{\pm0.4}$ | $64.0_{\pm0.4}$ | | | ✓ | 1 | 1 | $30.3_{\pm0.2}$ | $44.6_{\pm0.3}$ |
| | | ✓ | 10 | 1 | $51.2_{\pm0.4}$ | $66.2_{\pm0.2}$ | | | ✓ | 10 | 1 | $31.7_{\pm0.2}$ | $45.6_{\pm0.3}$ |
| | | ✓ | 1 | 4 | $50.6_{\pm0.3}$ | $65.3_{\pm0.2}$ | | | ✓ | 1 | 4 | $30.7_{\pm0.2}$ | $44.8_{\pm0.4}$ |

DD baseline and serves as a stable guidance for the generation of synthetic datasets. The whole optimizing target can be updated from Eq. (1) to:

$$\mathcal{S}^* = \arg\min_{\mathcal{S}} \mathcal{L}(\mathcal{S}, \mathcal{T}) + \alpha\mathcal{L}_{CLoM}, \tag{5}$$

where $\mathcal{L}_{CLoM} = \frac{1}{|\mathcal{S}|} \sum_{i=1}^{|\mathcal{S}|} \ell\left(f\left(\theta^*; s_i\right), y_i\right)$, $\ell$ is the cross-entropy loss, $\theta^*$ denotes the pre-trained model and $\alpha$ represents a tunable hyperparameter. In Section 6, we report the impact of different $\alpha$ on performance.

We implement CLoM on multiple state-of-the-art training pipelines of DD, including DC, DM, and DSA, to synthesize 10, and 50 images per class (IPC) respectively. For the sake of consistency, we opt for a pre-trained model ($\mathcal{N}_m = 1, \mathcal{N}_a = 1$) that has the same architecture (ConvNet-3) as the one used in DD. As shown in Table 1. We can observe that CLoM consistently improves the performance across all the baselines, which demonstrates that PTMs can indeed assist in DD.

## 5 EMPIRICAL STUDY

In the previous section, we have demonstrated that pre-trained models can assist in dataset distillation. Herein, we systematically study different options in PTMs, including initialization parameters, model architecture, training epoch and domain knowledge, and analyze the influence of each of these factors on synthetic datasets individually.

### 5.1 MODEL DIVERSIFICATION

First of all, we focus on model diversification, which includes two types: one is the diversification of initialization parameters, while the other is the diversification of model architecture. Herein, we use $\mathcal{N}_m$ to denote the number of PTMs (these PTMs are initialized with different random seeds and are thoroughly trained on the original dataset) and $\mathcal{N}_a$ to denote the number of different model architectures. To facilitate comparison, we set the experiments in Section 4 ($\mathcal{N}_m = 1, \mathcal{N}_a = 1$) as the control group and alter only one control variable at a time.

**Initialization parameters.** In Section 4, we utilize a single pre-trained model ($\mathcal{N}_m = 1, \mathcal{N}_a = 1$) with the same architecture as the one used in baselines to conduct experiments. In order to achieve diversification of initialization parameters, we train 10 models initialized with different random seeds from scratch ($\mathcal{N}_m = 10, \mathcal{N}_a = 1$). In each iteration, according to Eq. (5), the synthetic dataset is updated using one model at a time. As depicted in Table 1, when compared to a single model architecture and a single initialization parameter, introducing diverse initialization parameters leads to further performance improvements on synthetic datasets. These improvements can be attributed to

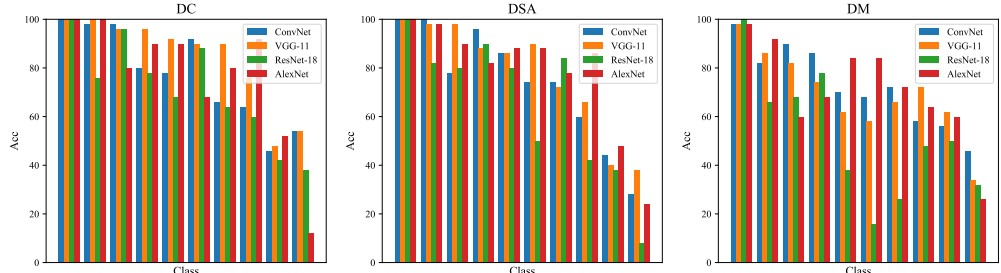

Figure 1: Classification accuracy of different architectures on various categories of synthetic datasets.

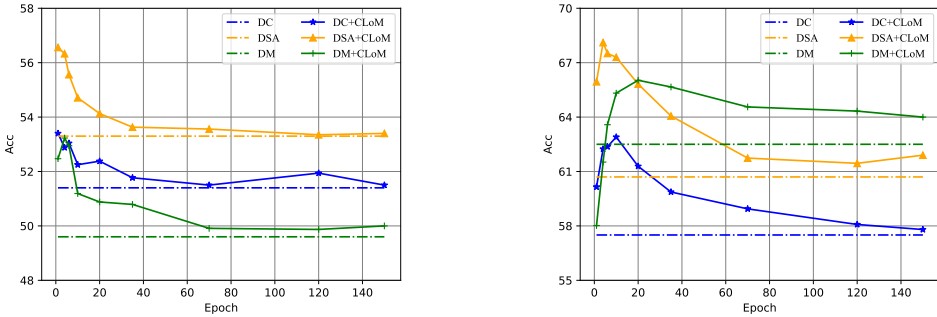

Figure 2: Performance of different baseline methods of DD. The models used in CLoM are derived from different stages of training. The dotted line represents the baseline.

model ensemble (Sagi & Rokach, 2018; Dong et al., 2020; Zhou et al., 2021) (two heads are better than one). Besides, we conduct an ablation study on $\mathcal{N}_m$ in Section 6.

**Model architecture.** Different from diverse initialization parameters, here, we use various model architectures. Before starting experiments, we evaluate the classification accuracy of different architectures on various categories of synthetic datasets, with the results in Fig. 1. We observe that different architecture excels in specific categories, allowing them to complement each other effectively. This finding highlights the rationality of utilizing diverse model architectures. Subsequently, we expand PTMs from a single architecture (ConvNet-3) to 4 architectures ($\mathcal{N}_m = 1, \mathcal{N}_a = 4$), including ConvNet-3 (Gidaris & Komodakis, 2018), AlexNet (Krizhevsky et al., 2017), VGG-11 (Simonyan & Zisserman, 2014) and ResNet-18 (He et al., 2016). These model architectures are commonly used in mainstream DD methods (Cui et al., 2022). At each iteration, we randomly select one pre-trained model from any of the 4 different architectures to calculate CLoM and update the synthetic dataset according to Eq. (5). We report the results in Table 1. Compared with a single model architecture and a single initialization parameter, diverse model architectures also enhance the performance of synthetic datasets to a certain extent. The ablation study on $\mathcal{N}_a$ is in Section 6.

## 5.2 TRAINING EPOCH

In previous studies, we have demonstrated that PTMs can assist in DD, and increasing the diversity of PTMs can further improve the performance of synthetic datasets. Herein, we want to ask *whether a (well-trained) PTM is necessary* and *if it's possible to use a sub-optimal model as a supervision signal instead*.

To answer the aforementioned questions, we conduct experiments using a single model architecture and a single initialization parameter ($\mathcal{N}_m = 1, \mathcal{N}_a = 1$). Different from using the well-trained PTM in Section 4, we utilize a sub-optimal model (ConvNet-3) instead. Specifically, we employ models at different training stages to calculate CLoM, encompassing epochs 1, 4, 6, 10, 20, 35, 70, 120, and 150. Subsequently, we apply these models to the baselines of DC, DSA and DM on CIFAR-10 with IPC=10 as well as IPC=50, and the results are presented in Fig. 2. Even with fewer training epochs, implementing CLoM on such sub-optimal models still leads to significant improvements, and in

some cases, it even outperforms well-trained PTMs. For example, on CIFAR-10 with IPC=50, DSA achieves an average accuracy of 68.1 at epoch 4, which is significantly higher than 61.9 achieved at epoch 150. This experimental result highlights that it is not necessary to utilize well-trained PTMs as supervision signals. Sub-optimal models can be employed instead, which not only enhances the performance of synthetic datasets but also takes less training cost.

## 5.3 DOMAIN KNOWLEDGE

In Section 5.2, we find that sub-optimal models can also assist in DD. Therefore, in this subsection, we further lower the threshold for PTMs. We wonder whether domain-specific PTMs are necessary as supervision signals. In other words, *are these PTMs, when utilized as supervision signals, limited to training on the original dataset, or can they also be trained on other datasets?* To this end, we utilize PTMs trained on other datasets to conduct experiments. However, a challenge arises when attempting to switch to PTMs trained on other datasets. Since the final classification layer of each PTM is tailored to the specific classes on which it was trained and there is no inherent correspondence between the original and new label sets, directly feeding the synthetic dataset into PTMs (trained on other datasets) for calculating CLoM is not feasible.

To tackle this issue of label mismatch, we introduce a novel loss term named **CCLoM**, which stands for **C**ontrastive **C**lassification **L**oss **o**f pre-trained **M**odel. Inspired by contrastive learning (Khosla et al., 2020; Jaiswal et al., 2020), this loss aims to minimize intra-class distances while maximizing inter-class distances using only feature representations. Specifically, in each training iteration, we sample a batch of examples $(x_\mathcal{B}, y_\mathcal{B})$ from the target dataset[5] $\mathcal{T}$ and feed them into the PTM (trained on the source dataset) to obtain real features $F(x_\mathcal{B})$. Here, $F$ is the feature extractor. Likewise, we take all examples $(x_\mathcal{S}, y_\mathcal{S})$ from the synthetic dataset $\mathcal{S}$ and feed them into the PTM to get synthetic features $F(x_\mathcal{S})$. We then utilize Eq. (6) to construct a distance matrix.

$$\mathcal{D}_{con} = 1 - \frac{F(x_\mathcal{B})^T F(x_\mathcal{S})}{\|F(x_\mathcal{B})\|_2 \|F(x_\mathcal{S})\|_2} \tag{6}$$

Due to the presence of labels, traditional contrastive losses are incapable of handling the case. In this case, we consider supervised contrastive learning (Khosla et al., 2020). Specifically, it contrasts all samples from the same class as positives against the negatives from the remainder of the batch. Let $O(y_\mathcal{B})$ and $O(y_\mathcal{S})$ be one-hot label encoding of the real batch and synthetic images, respectively. The label correspondence matrix is defined as $M = O(y_\mathcal{B})^T O(y_\mathcal{S})$, and CCLoM is computed as follows:

$$\mathcal{L}_{CCLoM} = \sum_{i=1}^{N} \frac{\sum \mathcal{D}_{con} \odot M}{\sum \mathcal{D}_{con}}, \tag{7}$$

where $N$ denotes the total batch of the target dataset. Finally, the whole optimizing target can be updated to Eq. (8) and Algorithm 1 in appendix provides the details of the calculation of CCLoM.

$$\mathcal{S}^* = \arg\min_{\mathcal{S}} \mathcal{L}(\mathcal{S}, \mathcal{T}) + \alpha \mathcal{L}_{CCLoM}. \tag{8}$$

We conduct evaluations on two image classification benchmarks: CIFAR-10 and PathMNIST. To control variables, we use a single model architecture and a single initialization parameter ($\mathcal{N}_m = 1$, $\mathcal{N}_a = 1$). The only difference from Section 4 is that the PTM is trained on ImageNet-32 or CIFAR-100. Table 2 reports the results, from which we can draw two key conclusion:

1) As supervision signals, PTMs do not need to be trained on the original dataset, they can also be trained on other datasets. This is evidenced by the consistent improvement of synthetic datasets with the assistance of CCLoM.

2) When domain-specific PTMs are not available, as supervision signals, domain-related datasets are preferred. It is worth noting that PathMNIST is a biomedical dataset and ImageNet-32 is a downsampled version of the original ImageNet. As supervision signals, PTMs trained on CIFAR-100 lead to significant performance gains (the average gain is 1.35) when CIFAR-10 is used as the target dataset for DD. Conversely, when the target dataset is switched to PathMNIST, the performance

---

[5]For ease of understanding, we refer to the dataset that needs to be distilled as the target dataset, and the dataset where PTMs are located as the source dataset in Section 5.3.

Table 2: Performance comparison to different DD methods. C10, C100, I32 and PM represents CIFAR-10, CIFAR-100, ImageNet-32 and PathMNIST, respectively. In this context, C100→C10 denotes using a PTM trained on CIFAR-100 as a supervision signal to guide the generation of synthetic datasets.

| Dataset | Method | IPC | | Avg Gain | Dataset | Method | IPC | | Avg Gain |
|---|---|---|---|---|---|---|---|---|---|
| | | 10 | 50 | | | | 10 | 50 | |
| C100→C10 | DC | $51.4_{\pm0.3}$ | $57.5_{\pm0.2}$ | +1.35 | C100→PM | DC | $56.1_{\pm1.2}$ | $69.3_{\pm0.6}$ | +0.15 |
| | DC+CCLoM | $52.5_{\pm0.4}$ | $59.5_{\pm0.2}$ | | | DC+CCLoM | $56.4_{\pm0.7}$ | $70.3_{\pm1.2}$ | |
| | Gain | +1.1 | +2.0 | | | Gain | +0.3 | +1.0 | |
| | DSA | $53.3_{\pm0.2}$ | $60.7_{\pm0.5}$ | | | DSA | $62.0_{\pm0.3}$ | $78.7_{\pm0.6}$ | |
| | DSA+CCLoM | $54.3_{\pm0.3}$ | $62.9_{\pm0.5}$ | | | DSA+CCLoM | $62.1_{\pm1.3}$ | $77.9_{\pm0.8}$ | |
| | Gain | +1.0 | +2.2 | | | Gain | +0.1 | -0.8 | |
| | DM | $49.6_{\pm0.3}$ | $62.5_{\pm0.7}$ | | | DM | $67.9_{\pm0.9}$ | $78.9_{\pm0.1}$ | |
| | DM+CCLoM | $50.0_{\pm0.3}$ | $63.9_{\pm0.5}$ | | | DM+CCLoM | $67.9_{\pm1.1}$ | $79.2_{\pm0.8}$ | |
| | Gain | +0.4 | +1.4 | | | Gain | 0.0 | +0.3 | |
| I32→C10 | DC | $51.4_{\pm0.3}$ | $57.5_{\pm0.2}$ | +0.83 | I32→PM | DC | $56.1_{\pm1.2}$ | $69.3_{\pm0.6}$ | +0.40 |
| | DC+CCLoM | $52.2_{\pm0.4}$ | $58.1_{\pm0.3}$ | | | DC+CCLoM | $56.5_{\pm0.8}$ | $70.0_{\pm0.6}$ | |
| | Gain | +0.8 | +0.6 | | | Gain | +0.4 | +0.7 | |
| | DSA | $53.3_{\pm0.2}$ | $60.7_{\pm0.5}$ | | | DSA | $62.0_{\pm0.3}$ | $78.7_{\pm0.6}$ | |
| | DSA+CCLoM | $53.6_{\pm0.2}$ | $61.9_{\pm0.1}$ | | | DSA+CCLoM | $63.0_{\pm0.8}$ | $78.7_{\pm1.3}$ | |
| | Gain | +0.3 | +1.2 | | | Gain | +1.0 | +0.0 | |
| | DM | $49.6_{\pm0.3}$ | $62.5_{\pm0.7}$ | | | DM | $67.9_{\pm0.9}$ | $78.9_{\pm0.1}$ | |
| | DM+CCLoM | $50.6_{\pm0.2}$ | $63.6_{\pm0.4}$ | | | DM+CCLoM | $68.7_{\pm0.6}$ | $78.4_{\pm0.6}$ | |
| | Gain | +1.0 | +1.1 | | | Gain | +0.8 | -0.5 | |

Table 3: Performance comparison to DM. Since the CLIP model is trained on a dataset of 400 million (image, text) pairs collected from the Internet, we call the dataset Internet.

| Dataset | Method | IPC | | Avg Gain | Dataset | Method | IPC | | Avg Gain |
|---|---|---|---|---|---|---|---|---|---|
| | | 1 | 10 | | | | 1 | 10 | |
| ImageNet→ImageNette | DM | $28.3_{\pm1.5}$ | $48.5_{\pm0.7}$ | +1.35 | ImageNet→ImageFruit | DM | $21.3_{\pm1.0}$ | $28.7_{\pm1.2}$ | +0.65 |
| | DM+CCLoM | $30.0_{\pm0.9}$ | $49.5_{\pm0.4}$ | | | DM+CCLoM | $22.1_{\pm0.3}$ | $29.2_{\pm1.0}$ | |
| | Gain | +1.7 | +1.0 | | | Gain | +0.8 | +0.5 | |
| Internet→ImageNette | DM | $28.3_{\pm1.5}$ | $48.5_{\pm0.7}$ | +1.20 | Internet→ImageFruit | DM | $21.3_{\pm1.0}$ | $28.7_{\pm1.2}$ | +1.30 |
| | DM+CCLoM | $31.2_{\pm0.7}$ | $49.0_{\pm0.3}$ | | | DM+CCLoM | $22.3_{\pm1.0}$ | $30.3_{\pm1.9}$ | |
| | Gain | +1.9 | +0.5 | | | Gain | +1.0 | +1.6 | |

improvement is minimal (the average gain is 0.15). We believe this is attributed to the significant domain gap (Nam et al., 2021) between CIFAR-100 and PathMNIST. The same phenomenon occurs when ImageNet-32 is utilized as the source dataset.

To further demonstrate that domain-specific PTMs are not necessary for supervision signals to provide precise guidance to DD, we conduct experiments on ImageNette and ImageFruit ($224 \times 224$ resolution scale). We utilize a pre-trained ResNet-18 (He et al., 2016) on ImageNet (Deng et al., 2009) and a pre-trained CLIP (Radford et al., 2021) model as supervision signals[6]. It's worth noting that these models can be obtained directly without the need for training. Then we conduct evaluations on DM with a batchsize of 512. The reason for using only DM is that DC and DSA require saving the computational graph to calculate the matching loss, which prevents them from conducting experiments on high-resolution datasets. Table 3 reports the results. The conclusion that domain-specific PTMs are not necessary for DD still holds true in the case of high resolution. More importantly, based on the success of the CLIP model in guiding ImageNette and ImageFruit synthesis, we are inspired to use existing foundational models, such as GPT-4 (OpenAI, 2023), to assist in DD.

## 5.4 VALIDATION AND EXTENSIONS

In the previous subsections, we analyze a variety of options in PTMs, including initialization parameters, model architecture, training epoch and domain knowledge, one by one. Herein, we choose various combinations of these options to perform extensive experiments. Since existing DD methods generally suffer from architecture overfitting (Zhao & Bilen, 2021; Cazenavette et al., 2022; Zhao & Bilen, 2023; Lei & Tao, 2023). That is, the optimization of synthetic dataset is closely tied to

---

[6]ResNet-18 is downloaded from torchvision. As for the CLIP model, we use ResNet-50 as the base architecture for the image encoder.

Table 4: Cross-architecture performance comparison to different baseline methods of DD. The distillation architecture is ConvNet-3. The experiments are conducted on CIFAR-10 with IPC-50. WT and SO represent well-trained and sub-optimal, respectively. **Bold entries** are best results.

| Method | CLoM | $\mathcal{N}_m$ | $\mathcal{N}_a$ | Training Epoch | Model | | | | Avg Gain |
| --- | --- | --- | --- | --- | --- | --- | --- | --- | --- |
| | | | | | ConvNet-3 | VGG-11 | ResNet-18 | AlexNet | |
| DC | ✗ | - | - | - | $57.5_{\pm0.2}$ | $50.4_{\pm0.9}$ | $46.7_{\pm0.4}$ | $37.6_{\pm0.6}$ | - |
| | ✓ | 10 | 1 | WT | $60.2_{\pm0.3}$ | $52.9_{\pm0.7}$ | $47.4_{\pm0.3}$ | $37.7_{\pm0.4}$ | +1.5 |
| | ✓ | 10 | 4 | WT | $61.3_{\pm0.2}$ | $55.2_{\pm0.7}$ | $48.6_{\pm1.1}$ | $\mathbf{37.9}_{\pm0.4}$ | +2.7 |
| | ✓ | 10 | 4 | SO | $\mathbf{63.0}_{\pm0.2}$ | $\mathbf{59.2}_{\pm0.6}$ | $\mathbf{50.3}_{\pm1.2}$ | $37.8_{\pm1.0}$ | **+4.5** |
| DSA | ✗ | - | - | - | $60.7_{\pm0.5}$ | $50.9_{\pm0.7}$ | $50.3_{\pm1.1}$ | $46.0_{\pm0.3}$ | - |
| | ✓ | 10 | 1 | WT | $65.1_{\pm0.5}$ | $52.8_{\pm0.9}$ | $51.6_{\pm0.8}$ | $45.5_{\pm0.5}$ | +1.8 |
| | ✓ | 10 | 4 | WT | $66.4_{\pm0.3}$ | $55.9_{\pm0.9}$ | $52.7_{\pm0.8}$ | $46.1_{\pm0.9}$ | +3.3 |
| | ✓ | 10 | 4 | SO | $\mathbf{69.2}_{\pm0.2}$ | $\mathbf{61.7}_{\pm0.7}$ | $\mathbf{60.7}_{\pm1.0}$ | $\mathbf{58.9}_{\pm0.7}$ | **+10.7** |
| DM | ✗ | - | - | - | $62.5_{\pm0.4}$ | $55.9_{\pm0.5}$ | $54.6_{\pm0.5}$ | $50.6_{\pm0.7}$ | - |
| | ✓ | 10 | 1 | WT | $66.2_{\pm0.2}$ | $59.6_{\pm0.4}$ | $56.9_{\pm0.6}$ | $52.4_{\pm0.6}$ | +2.9 |
| | ✓ | 10 | 4 | WT | $66.4_{\pm0.3}$ | $60.4_{\pm0.4}$ | $58.1_{\pm0.8}$ | $\mathbf{57.1}_{\pm0.9}$ | +4.6 |
| | ✓ | 10 | 4 | SO | $\mathbf{67.5}_{\pm0.1}$ | $\mathbf{62.1}_{\pm0.6}$ | $58.3_{\pm1.0}$ | $56.3_{\pm1.1}$ | **+5.2** |

the specific model architecture, and there is a performance drop when the synthetic dataset is applied to unknown architectures. Therefore, we employ CLoM with various combinations of options to mitigate architecture overfitting in DD. Specifically, we utilize ConvNet-3 with the default hyperparameters from the previous work (Cui et al., 2022) to generate synthetic datasets. Subsequently, we evaluate the performance of synthetic datasets on ConvNet-3, VGG-11, ResNet-18 and AlexNet. With the goal of conducting a fair comparison, we utilize the same training configurations for evaluation. All experiments are conducted on CIFAR-10 with IPC-50.

Table 4 exhibits the experimental results. In general, different combinations of options mitigate architecture overfitting to varying degrees. Among these combinations, sub-optimal models (According to Fig. 2, we select the models at the epoch with the best performance as sub-optimal models.) with diverse initialization parameters and various model architectures ($\mathcal{N}_m = 10, \mathcal{N}_a = 4$) perform best. Specifically, among these model architectures, DSA achieves an average performance improvement of 10.7% on synthetic datasets with the assistance of CLoM. In comparison, the improvements on DC and DM are relatively small but still substantial, at 4.5% and 5.2%, respectively. Besides, we visualize the feature distribution of these synthetic images and compare them to the feature distribution learned by DC, DSA and DM. Following the settings of Zhao & Bilen (2023), we utilize a model trained on the whole training set to extract features and visualize the features with t-SNE (Van der Maaten & Hinton, 2008). As depicted in Fig. A1, compared to the baseline, our synthetic images more accurately reflect the distribution of the original training set.

For additional supplementary experiments, please refer to Appendix A.

# 6 ABLATION STUDY

In Section 5.1, we find that diverse initialization parameters and diverse model architectures can enhance the performance of synthetic datasets. Here, we conduct ablation studies on $\mathcal{N}_m$ and $\mathcal{N}_a$. Additionally, we also investigate the influence of $\alpha$ on the performance of synthetic datasets. Unless otherwise specified, all ablation studies are conducted on CIFAR-10 with IPC=50.

**Ablation on $\alpha$.** We conduct experiments on the basis of Section 4 with varying $\alpha$. The outcome is presented in the left part of Fig. 3. We observe that different values of $\alpha$ have a slight impact on the performance of the synthetic dataset. This impact is primarily attributed to the trade-off between two losses in Eq. (5).

**Ablation on $\mathcal{N}_m$.** Following the experimental settings of Section 4, we use different $\mathcal{N}_m$ to perform experiments. As shown in the right part of Fig. 3, with the number of $\mathcal{N}_m$ increasing, the overall performance of the synthetic dataset exhibits an upward trend. The best results are achieved when $\mathcal{N}_m = 10$. Hence, in the previous experiments, we set $\mathcal{N}_m = 10$ by default.

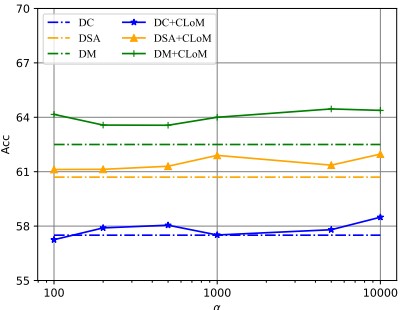 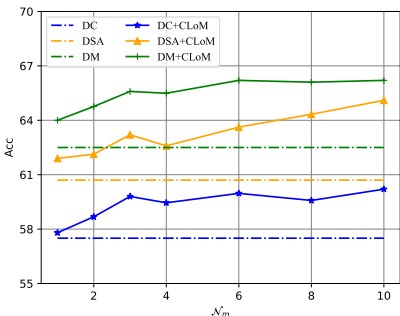

Figure 3: Ablation study on $\alpha$ and $\mathcal{N}_m$. The dotted line represents the baseline.

Table 5: Explorations of the diversity of model architectures. This experiment is conducted based on DC. **Bold entries** are best results.

| ConvNet-3 | VGG-11 | ResNet-18 | AlexNet | ConvNet-4 | Performance |
|:---------:|:------:|:---------:|:-------:|:---------:|:-----------:|
| ✓ | | | | | $57.8_{\pm 0.3}$ |
| | ✓ | | | | $57.7_{\pm 0.3}$ |
| | | ✓ | | | $57.3_{\pm 0.3}$ |
| | | | ✓ | | $57.3_{\pm 0.6}$ |
| | | | | ✓ | $58.2_{\pm 0.2}$ |
| ✓ | ✓ | | | | $58.7_{\pm 0.5}$ |
| ✓ | ✓ | ✓ | | | $59.1_{\pm 0.6}$ |
| ✓ | ✓ | ✓ | ✓ | | $59.7_{\pm 0.6}$ |
| ✓ | ✓ | ✓ | ✓ | ✓ | $\mathbf{60.0}_{\pm 0.2}$ |

**Ablation on $\mathcal{N}_a$.** Similarly, based on the experiment settings of Section 4, we only change $\mathcal{N}_a$. Our experiments cover a range of scenarios, from a single model architecture to various combinations of model architectures. To make the experiment more convincing, we further add the ConvNet-4. The results are exhibited in the Table 5. We observe that diverse model architectures consistently lead to improved performance of the synthetic dataset. The best results are achieved with all five model architectures included.

## 7 CONCLUSION

In this paper, we explore the question: "Can pre-trained models assist in dataset distillation?" To this end, we introduce two plug-and-play loss terms, *i.e.*, CLoM and CCLoM, portable to any existing DD baseline. By leveraging PTMs as supervision signals, they serve as stable guidance on the optimization direction of synthetic datasets. Then we systematically study the effects of different options in PTMs on DD, including initialization parameters, model architecture, training epoch and domain knowledge, and summarize several key findings. Based on these findings, we achieve significant improvements in cross-architecture generalization compared to the baseline method. Finally, we aspire that our research will facilitate the DD community to concentrate on PTMs and inspire the development of innovative algorithms in the future.

**Limitations.** While PTMs as supervision signals provide stable guidance for optimizing synthetic datasets, there is a potential limitation that should be acknowledged. For example, the training cost of PTMs cannot be overlooked. This can be mitigated by using sub-optimal models (see Section 5.2). Besides, thanks to the CCLoM, domain-agnostic PTMs are able to serve as supervision signals (see Section 5.3), which eliminates the need to train PTMs.

**Future Works.** In this paper, we demonstrate that domain-agnostic PTMs can also assist in DD (see Section 5.3). Therefore, a promising direction is harnessing existing foundational models, such as CLIP (Radford et al., 2021) and GPT-4 (OpenAI, 2023), to steer and enhance DD. Thanks to the generalization capabilities exhibited by these foundational models, the application of DD in more intricate scenarios, such as semantic segmentation (Long et al., 2015), objective detection (Girshick et al., 2014) and multi-modal (Radford et al., 2021), becomes feasible.

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

**Algorithm 1** PyTorch-like pseudo-code for CCLoM.

```
# num_classes: number of classes
# batch_size: batch size of real images

def calculating_CCLoM(real_img, real_label, syn_img, syn_label):
    # One-Hot Encoding to real label
    real_label_onehot = torch.zeros((batch_size, num_classes))
    real_label_onehot[torch.arange(batch_size), real_label] = 1

    # One-Hot Encoding to synthetic label
    syn_label_onehot = torch.zeros((ipc * num_classes, num_classes))
    syn_label_onehot[torch.arange(ipc * num_classes), syn_label] = 1

    # True and false sample matrix
    labels = syn_label_onehot @ real_label_onehot.T

    # extract feature representations
    real_feat = model(real_img)
    syn_feat = model(syn_img)
    real_feature_norm = real_feat / real_feat.norm(dim=-1, keepdim=True)
    syn_feature_norm = syn_feat / syn_feat.norm(dim=-1, keepdim=True)

    cos_dist = 1 - syn_feature_norm @ real_feature_norm.T
    cclom = torch.sum(labels * cos_dist) / torch.sum(cos_dist)

    return cclom
```

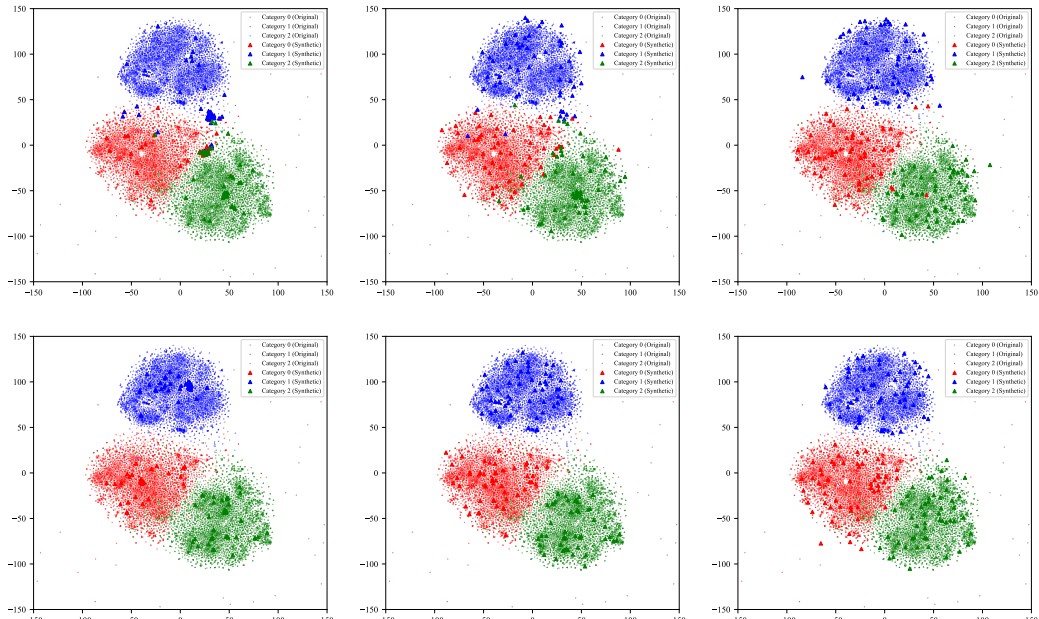

Figure A1: Distributions of synthetic images learned by DC, DSA and DM. The first row represents the distribution without CLoM, while the second row represents the distribution with CLoM using the best combination of options. From left to right are the distributions of DC, DSA, and DM. Best viewed in color.

# A  APPENDIX

We apply the proposed CLoM to state-of-the-art methods, including IDC (Kim et al., 2022), DiM (Wang et al., 2023), DREAM (Liu et al., 2023) and TM (Cazenavette et al., 2022), and compare the performance in Table A1. For fair comparison, the experiments are conducted under the default parameters provided in original papers. Herein, we utilize well-trained models with diverse initialization parameters and single model architectures ($\mathcal{N}_m = 10$, $\mathcal{N}_a = 1$) to conduct experiments. The stable performance improvements demonstrate that even with state-of-the-art DD methods, PTMs can still improve the performance of synthetic datasets.

Table A1: The performance comparison to state-of-the-art methods on CIFAR-10. Underline denotes results from the original papers. **Bold entries** are best results.

| Method | IPC | CLoM | | Method | IPC | CLoM | |
|---|---|---|---|---|---|---|---|
| | | ✗ | ✓ | | | ✗ | ✓ |
| IDC | 10 | $\underline{67.5_{\pm 0.5}}$ | $\mathbf{69.4}_{\pm 0.3}$ | DREAM | 10 | $\underline{69.4_{\pm 0.4}}$ | $\mathbf{70.0}_{\pm 0.4}$ |
| | 50 | $\underline{74.5_{\pm 0.1}}$ | $\mathbf{76.3}_{\pm 0.2}$ | | 50 | $\underline{74.8_{\pm 0.1}}$ | $\mathbf{76.1}_{\pm 0.1}$ |
| DiM | 10 | $\underline{66.2_{\pm 0.5}}$ | $\mathbf{67.3}_{\pm 0.5}$ | TM | 10 | $\underline{65.3_{\pm 0.7}}$ | $\mathbf{66.3}_{\pm 0.4}$ |
| | 50 | $\underline{72.6_{\pm 0.4}}$ | $\mathbf{72.7}_{\pm 0.3}$ | | 50 | $\underline{71.6_{\pm 0.2}}$ | $\mathbf{73.1}_{\pm 0.2}$ |

