# OpenReview forum: "Can pre-trained models assist in dataset distillation?"
_ICLR.cc/2024/Conference — ICLR 2024 Conference Withdrawn Submission_

### Official Review · Reviewer_nf4G · 2023-10-31

**Soundness:** 2 fair
**Presentation:** 3 good
**Contribution:** 2 fair
**Rating:** 5
**Confidence:** 2

**Summary:**

This paper investigates to leverage pretrained models for providing extra supervision signals for data distillation. For the domain-specific case where models are pre-trained on the target dataset, a loss term called CoLM is introduced. For the domain-agnostic case where the model is pre-trained on another dataset, a loss term called CCoLM is introduced. Experiments with various options (including initialization parameters, model architecture, training epoch and domain knowledge) are conducted over 3 representative DD methods (DC, DSA and DM) and verify the effectiveness of both domain-specific and domain-agnostic cases.

**Strengths:**

1. The writing is good and the paper is easy to follow
2. The idea of leveraging PTMs in DD is well motivated and novel
3. The study of various options is impressive

**Weaknesses:**

1. Lack of analysis of the training time/storage cost of leveraging PTMs in DD, especially the domain-specific case. Pretraining models with different initialization and model architecture over the target dataset seems time-consuming (even with early-stopping).
2. The analysis of cross-architecture generalization. The reported number in table 4 seems inconsistent with that in table 3 of the original DM paper.

**Questions:**

How it performs when using only CLoM (i.e., $\alpha \to \infty$) for DD (both domain-specific and domain-agnostic case)?

---

### Official Review · Reviewer_qWjp · 2023-11-01

**Soundness:** 2 fair
**Presentation:** 2 fair
**Contribution:** 2 fair
**Rating:** 5
**Confidence:** 5

**Summary:**

This paper study how a pretrained model can help dataset distillation (DC, DSA and DM) from many perspectives, like model diversity, training epochs, etc, and then proposed two losses to involve the pretrained models into the distillation process. The authors show those approaches can further improve the performance of dataset distillation.

**Strengths:**

1. This paper studies how the pre-trained models can help the dataset distillation from different perspectives.
2. This paper proposed two additional loss terms (CLoM and CCLoM) to incorporate PTM in the process.

**Weaknesses:**

1. The model diversification and whether or not the models need to be well-trained have been studied in CVPR 2023 paper, "Accelerating Dataset Distillation via Model Augmentation". This paper pointed out these two points in their paper.
2. The improvement with CCLoM (Table 2) is very marginal, the average gain only up to 1.35%.

**Questions:**

1. Could the authors elaborate more what is the difference from MTT? Those PTM are the same to the expert models.

2. CCLoM can be also applied to the PTM trained on the same dataset by simply using the feature representation layer, this should be an ablation to validate CLoM is more effective.

3. What is the training overhead of using PTM? What does it affect the training time for dataset distillation?

4. After getting N_m models with different seeds, is there any way to quantify the diversity of models and then is it possible to correlate the diversity to the performance gain in distillation?

---

### Official Review · Reviewer_Kiov · 2023-11-01

**Soundness:** 1 poor
**Presentation:** 2 fair
**Contribution:** 1 poor
**Rating:** 3
**Confidence:** 4

**Summary:**

This paper studies whether Pre-trained Models (PTMs), which contain extensive knowledge from the original dataset, can assist with dataset distillation. Through experiments, it's revealed that model diversity and varying initialization parameters can enhance DD, sub-optimal models might sometimes outperform well-trained ones, and while PTMs don't necessarily need to come from the same domain, a close domain match is beneficial. The research introduces loss terms CLoM and CCLoM to guide synthetic dataset distillation.

**Strengths:**

This paper is well-motivated and they study an important problem.

**Weaknesses:**

1. A primary concern is the seemingly ambiguous use of the concept of Pre-trained Models (PTMs). There are very limited details on the data that are used for PTMs during the pretraining stage. The introduction and related work sections give the impression that pretraining follows the convention of using the ImageNet dataset, similar to methods they cited. However, I realize until section 5.2 when the discussion on pretraining across various epochs (1, 4, 6, 10...) is provided, that this is probably not pretrained on the imagenet dataset. However, if the model is pretrained on CIFAR10 (that further used for distillation), one would expect it to naturally have a better performance during distillation. How is it different from the buffer training stage in previous works e.g. [1,2]?
2. The overall structure and writing of this paper lack clarity and cohesiveness. For example, the decision to space out section 3.2 (experiment setup) and section 5, and to introduce losses in sections 4 and 5.3 separately, results in disjointed reading and necessitates excessive cross-referencing.
3. Two of the highlighted key findings do not appear particularly new. The model diversification is a pretty established technique that are widely used in a lot of DD works, and domain knowledge part also seems to be a widely known concept. The point is, I do appreciate the authors provided very detailed experiments but I don't think the authors communicate their key findings very effectively in the introduction part.
4. The paper has a lot of technique details missing e.g. dataset resolution, full dataset training performance, and specifics about datasets used for model pretraining.
5. For the distilled dataset evaluation using data augmentation techniques. What's the size of the final distilled-then-augmented dataset?  Similarly, as mentioned in point 4, there's an absence of full dataset training performance metrics, especially when data augmentation techniques are applied, which makes it hard to evaluate the results.
6. In Figure 1, on which dataset was this experiment conducted? It is not identified or described within the paper.


[1] Cazenavette, George, et al. "Generalizing Dataset Distillation via Deep Generative Prior." Proceedings of the IEEE/CVF Conference on Computer Vision and Pattern Recognition. 2023.

[2] Zhou, Yongchao, Ehsan Nezhadarya, and Jimmy Ba. "Dataset distillation using neural feature regression." Advances in Neural Information Processing Systems 35 (2022): 9813-9827.

**Questions:**

Same as weaknesses section, my biggest concern is related to the concept of the pre-trained models.

---

### Official Review · Reviewer_KYoU · 2023-11-03

**Soundness:** 2 fair
**Presentation:** 2 fair
**Contribution:** 2 fair
**Rating:** 6
**Confidence:** 4

**Summary:**

This paper investigates the role of pre-trained models (PTMs) in dataset distillation (DD) and explores whether PTMs can assist in improving the performance of synthetic datasets. The authors introduce two plug-and-play loss terms, Classification Loss of pre-trained Model (CLoM) and Contrastive Classification Loss of pre-trained Model (CCLoM), which leverage PTMs as supervision signals to guide the optimization of synthetic datasets. They systematically study the effects of different options in PTMs, including initialization parameters, model architecture, training epoch, and domain knowledge, and summarize several key findings. The experimental results show that PTMs can significantly enhance cross-architecture generalization in DD. Additionally, the authors demonstrate that sub-optimal models can also be used as supervision signals and that domain-agnostic PTMs can provide precise guidance to DD. The paper concludes by discussing the limitations and potential future directions of this research.

**Strengths:**

1.	The paper systematically studies different options in pre-trained models (PTMs), such as initialization parameters, model architecture, training epoch, and domain knowledge. And the influence of hyperparameters is also considered. This comprehensive analysis helps in understanding the influence of each factor on synthetic datasets individually.
2.	The paper presents the findings and analysis in a clear and organized manner, making it easy for readers to follow and understand the experiments and results.
3.	The paper’s focus on dataset distillation and the analysis of factors influencing synthetic datasets has practical implications in the field of machine learning and data preprocessing. It provides insights into how to effectively utilize pre-trained models for improving dataset quality.

**Weaknesses:**

The paper primarily focuses on empirical studies and experimental results. It would be valuable to provide more theoretical analysis and insights into why the proposed approach works, such as discussing the underlying principles or mathematical foundations.

This paper investigates the application of pre-trained models (PTMs) in dataset distillation (DD) and their potential to enhance the performance of synthetic datasets. The authors introduce two novel loss terms, CLoM and CCLoM, to leverage PTMs as guidance for optimizing synthetic datasets. The paper systematically explores various aspects of PTMs, including initialization parameters, model architecture, training epoch, and domain knowledge, offering valuable insights.
The comprehensive analysis of PTM options and their impact on synthetic datasets provides a thorough understanding of the individual factors' influence. The presentation of findings and analysis is clear and well-organized, making it accessible for readers to follow and grasp the experimental results. Furthermore, the paper's focus on dataset distillation and the exploration of factors influencing synthetic datasets has practical implications in the machine learning and data preprocessing domain, offering insights into effectively leveraging PTMs for dataset enhancement.
However, the paper could benefit from a more in-depth theoretical analysis. While the empirical studies and experimental results are valuable, providing a deeper understanding of why the proposed approach is effective would enhance the paper's quality. A discussion of the underlying principles or mathematical foundations would further strengthen the theoretical foundation of the research. Furthermore, the manuscript exhibits some minor imperfections, with an identified inaccuracy in Equation (1).

**Questions:**

Please see the weakness.